# Outcomes of Patients with Amputation following Electrical Burn Injuries

Eunyeop Kim [1], Bingchun Wan [2], Kyra Jeanine Solis-Beach [3] and Karen Kowalske [3,*]

1 University of Texas Southwestern Medical School, Dallas, TX 75390, USA; eunyeop.kim@utsouthwestern.edu
2 Department of Surgery, University of Texas Southwestern Medical Center, Dallas, TX 75390, USA; bingchun.wan@utsouthwestern.edu
3 Department of Physical Medicine and Rehabilitation, University of Texas Southwestern Medical Center, Dallas, TX 75390, USA; kyrajeanine.solis-beach@utsouthwestern.edu
* Correspondence: karen.kowalske@utsouthwestern.edu

**Abstract:** This study aimed to examine patients who sustained amputation as a result of electrical burns and to evaluate their long-term health outcomes compared to non-electrical burn patients with amputation. A retrospective analysis was conducted on burn patients from 1993 to 2021, utilizing the Burn Model System National Database, which includes the Veterans RAND 12-Item Health Survey and the Patient-Reported Outcomes Measurement Information System 29. The data was collected at discharge, 6 months, and 12 months after the burns occurred. The findings revealed that the rate of amputation was significantly higher in electrical burn patients (30.3%) compared to non-electrical burn patients (6.6%) ($p < 0.0001$). At the time of discharge, electrical burn patients with amputation exhibited significantly lower physical component scores (PCS = 34.00 ± 8.98) than electrical burn patients without amputation (PCS = 44.66 ± 9.90) ($p < 0.05$). However, there were no significant differences in mental component scores observed between patients, regardless of the burn type or amputation. Among all patient groups, non-electrical burn survivors with amputation faced the greatest challenges in terms of physical and social well-being, likely due to larger total body surface area burns. This study emphasizes the importance of early rehabilitation for electrical burn patients with amputation and highlights the need for ongoing support, both physically and socially, for non-electrical burn survivors with amputation. These findings, consistent with previous studies, underscore the necessity of providing psychological support to all burn survivors.

**Keywords:** electrical burns; amputation; burns; long-term outcomes

## 1. Introduction

Electrical burn injuries are prevalent worldwide, constituting a significant proportion of burn cases. These injuries account for approximately 4.5% of all burn injuries globally and can escalate to 27% of all burn injuries in developing countries [1]. The severity of electrical injuries is underscored by their high mortality rates, reaching up to 5.2% in cases involving high voltages [2]. Patients with electrical burns often experience a severe inflammatory response, which can lead to multiple organ failure in acute settings [2–5].

Moreover, electrical injuries pose unique challenges compared to burns from other etiologies, as the extent of injury can be particularly difficult to assess at initial presentation [6]. Complications caused by these injuries can appear up to five years after the initial incident, necessitating long-term monitoring to identify and manage symptoms [7,8]. Additionally, cardiovascular complications and peripheral neuropathy occur more frequently in electrical than in thermal burns. Similar to other extensive burn injuries, abnormal temperature sensation, sensory loss, and pruritis may continue for several years after the injury [9,10]. In addition, other complications, such as epilepsy, cataracts, tongue atrophy, and paraplegia, may also have an insidious onset [11,12].

Limb loss, a common complication of electrical burn injuries, substantially affects the lives of affected individuals [13]. However, the exact national incidence of amputation in electrical burn patients in the U.S. remains unclear. Survivors of electrical burns who undergo amputation often face severe social stigma and may require support for effective community and social reintegration [14,15]. Patients with amputation also face greater difficulties in returning to work, and they are also more likely to develop psychological difficulties, such as post-traumatic stress disorder, depression, and body image dissatisfaction [16,17].

Owing to the complications that lead to changes in function and quality of life, the long-term rehabilitation of patients with electrical burns is mandatory. Previous research [2,18–20] has investigated the effects of amputations in patients with burns and the long-term complications of electrical burn injuries; however, comprehensive studies specifically focused on electrical burn survivors with amputation using large national-scale data in the U.S. are lacking. Understanding the unique complications and outcomes experienced by electrical burn survivors with amputation can establish the groundwork for better care management.

Therefore, the objective of this study is to analyze the incidence of amputation in electrical burn injuries and investigate how it affects the long-term physical, mental, and social status of patients compared to those with burns resulting from other etiologies. By elucidating the specific challenges faced by electrical burn survivors with amputations, this research aims to contribute to the development of improved intervention and care plans that optimize their rehabilitation journey and enhance their overall well-being.

## 2. Materials and Methods

Retrospective reviews of burn patients were conducted using the Burn Model System (BMS) National Database, collected from 1993 to 2021 [21]. This study was exempt from the institutional review board approval because the information obtained from the database was a nationally available de-identified dataset. Burn survivors who had consented to the study, had documentation of burn etiology, and were alive at discharge were included in the study. Data were collected at discharge, 6 months post-burn, and 12 months post-burn to track the patients' progress over time.

To begin the analysis, the patients were first divided into two groups: patients with electrical burn and patients with non-electrical burn. Patients with electrical burn were defined as those with either a primary electrical etiology or exposure to high voltages/lightning and had undergone wound closure surgery according to the primary admission criteria. Patients with non-electrical burn etiologies were defined as burn patients who did not meet the above criteria, and involved etiologies including fire/flame, scald, contact with a hot object, grease, tar, chemical, hydrofluoric acid, UV light, flash burns, and other unknown burns. Furthermore, within each group, the patients were further divided into electrical burn patients with and without amputation, and non-electrical burn patients with and without amputation. Patients with unknown amputation status were excluded from the study. The location and level of amputation was not specified in the dataset and was thus excluded from the analysis.

The variables chosen for analysis included the physical component score (PCS) and mental component score (MCS) from the Veterans RAND 12-Item Health Survey (VR12) and social role score from Patient-Reported Outcomes Measurement Information System 29 (PROMIS29). VR12 and PROMIS29 are widely used questionnaires validated to assess physical and mental health status and to track changes over time [22,23]. VR12 is a self-administered survey consisting of 12 items that measure health-related quality of life. The 12 items result in the PCS and MCS, which describe physical and psychological health statuses, respectively [24]. The PCS and MCS scores range from 0 to 100, with a population mean of 50 and higher scores indicating a better physical or mental health status [23]. PROMIS29 is a self-report tool measuring seven health domains (physical function, fatigue, pain interference, depressive symptoms, anxiety, social participation, and

sleep disturbances) using four items per domain to assess health-related outcomes [22]. PROMIS29 is standardized to US population data, producing scores ranging from 0 to 100, with a population mean of 50 and a standard deviation of 10.

Statistical analyses were performed using SAS 9.4, and *p*-values of <0.05 were defined as statistically significant. A chi-square test was performed to compare the amputation frequency between the electrical and non-electrical etiology burn groups. Mann–Whitney U tests were used to compare the total body surface area (TBSA) (%) burnt between the electrical burn patients and non-electrical burn patients, and between electrical burn survivors with amputation and non-electrical burn survivors with amputation. The overall mean differences in the PROMIS29 and VR12 scores between the patient groups were analyzed using the Kruskal–Wallis test and post hoc analyses using Dunn–Bonferroni test to identify specific group differences.

## 3. Results

### 3.1. Demographics

The study sample consisted of 408 patients with electrical burns and 6341 patients with non-electrical etiology burns. The median age was not significantly different between electrical burn patients (28.44 years) and non-electrical burn patients (30 years). Significantly more patients with electrical burns were male (93.63% male, 6.37% female) compared to patients with non-electrical burns (69.41% male, 30.59% female) (*p*-value < 0.00001). Participants in the electrical burn group were significant more likely to be Hispanic (41.12%) compared to the non-electrical burn group (27.06%) (*p*-value < 0.0001). The results are displayed in Table 1.

**Table 1.** Comparisons of non-electrical burn and electrical burn group.

| Variable | Non-Electrical Burn | Electrical Burn | *p*-Value |
|---|---|---|---|
| Ethnicity | *n* (%) | *n* (%) | <0.0001 |
| Hispanic | 1632 (27.06) | 162 (41.12) | |
| Non-Hispanic | 4399 (72.94) | 232 (58.88) | |
| Gender | *n* (%) | *n* (%) | <0.0001 |
| Male | 4401 (69.41) | 382 (93.63) | |
| Female | 1940 (30.59) | 26 (6.37) | |
| Age | | | 0.8141 |
| Mean ± Std | 30.83 ± 21.81 | 29.91 ± 16.16 | |
| Median | 29.99 | 28.44 | |
| TBSA (%) burnt | | | 0.0027 |
| Mean ± Std | 23.56 ± 20.36 (%) | 20.44 ± 19.5 (%) | |

### 3.2. Amputation Rate and Total Body Surface Area (TBSA) (%) Burnt

Patients with electrical (30.3%) burns were more likely to undergo amputation than those with non-electrical burns (6.6%) (*p* < 0.0001).

The TBSA (%) burnt results are summarized in Figure 1. The average TBSA (%) burnt of the non-electrical burn group was significantly higher than that of electrical burn patients (23.6% vs. 20.4%; *p* = 0.0027). The average TBSA (%) burnt for non-electrical burn survivors with amputation was significantly higher than that of electrical burn survivors with amputation (41.5% vs. 21.1%; *p* < 0.0001).

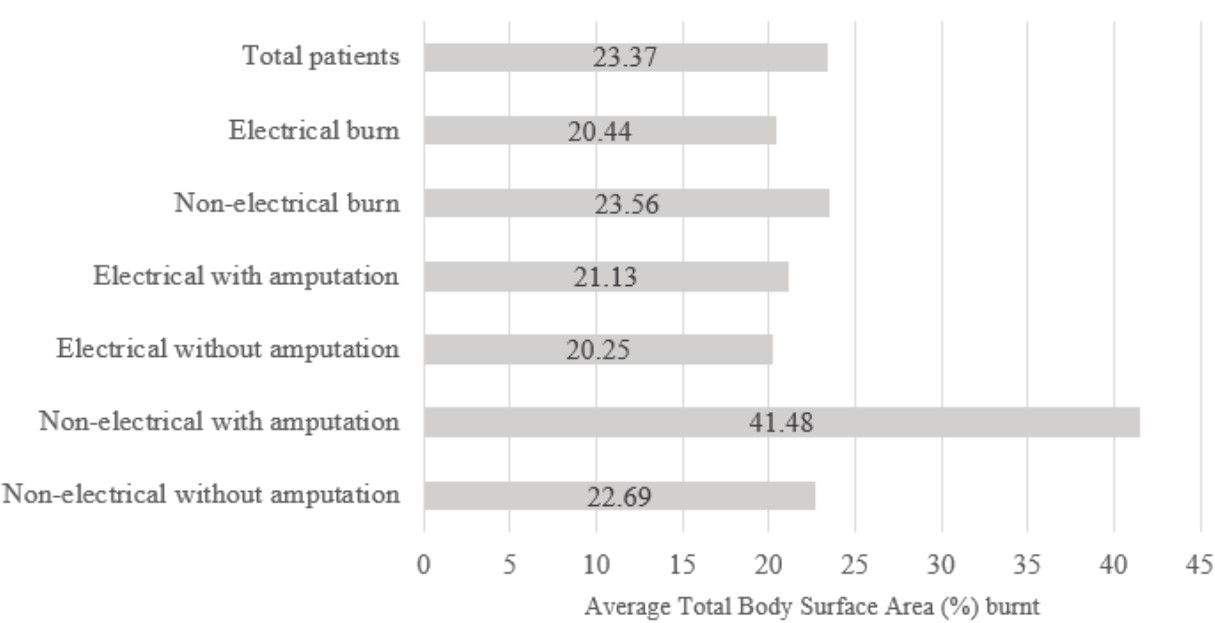

**Figure 1.** Average total body surface area (%) burnt.

### 3.3. Physical Component Score (PCS)

At discharge, the electrical burn with amputation group had lower PCSs than electrical burn survivors without amputation (34.00 vs. 44.66; $p$ = 0.03), and the non-electrical burn with amputation group had lower PCSs than the non-electrical burn without amputation group (35.90 vs. 43.44; $p$ = 0.0012). Non-electrical burn survivors with amputation reported significantly lower PCSs than non-electrical burn survivors without amputation at six months (38.6 vs. 45.2; $p$ = 0.0185), and one year post-injury (39.18 vs. 46.30; $p$ = 0.0040) as well. The results are summarized in Tables 2 and 3, and Figure 2.

**Table 2.** Physical component score (PCS) with Kruskal–Wallis test.

| | | | Month 0 | | | |
|---|---|---|---|---|---|---|
| **Variable** | **Statistics** | **Non-electrical amput** | **Non-electrical nonamput** | **Elec amput** | **Elec nonamput** | ***p*-value** |
| PCS | Mean ± SD | 35.90 ± 11.67 | 43.44 ± 10.71 | 34.00 ± 8.98 | 44.66 ± 9.90 | 0.0002 |
| | Median | 34 | 45 | 37 | 47 | |
| | *n* | 30 | 516 | 11 | 29 | |
| | | | **Month 6** | | | |
| **Variable** | **Statistics** | **Non-electrical amput** | **Non-electrical nonamput** | **Elec amput** | **Elec nonamput** | ***p*-value** |
| PCS | Mean ± SD | 38.69 ± 10.56 | 45.26 ± 11.08 | 37.67 ± 9.97 | 45.31 ± 9.09 | 0.0036 |
| | Median | 39 | 47 | 38 | 46 | |
| | *n* | 26 | 452 | 9 | 26 | |
| | | | **Month 12** | | | |
| **Variable** | **Statistics** | **Non-electrical amput** | **Non-electrical nonamput** | **Elec amput** | **Elec nonamput** | ***p*-value** |
| PCS | Mean ± SD | 39.18 ± 9.65 | 46.30 ± 10.67 | 40.11 ± 12.22 | 47.70 ± 10.20 | 0.0014 |
| | Median | 40 | 48 | 39 | 50 | |
| | *n* | 28 | 400 | 9 | 23 | |

**Table 3.** Physical component score post hoc analysis with Dunn–Bonferroni test.

| Month 0 | | | |
|---|---|---|---|
| Group comparison | Difference between means | Simultaneous 95% confidence limits | *p*-value |
| Non-electrical amput vs. Non-electrical nonamput | −7.5399 | −12.8583, −2.2215 | 0.0012 |
| Elec amput vs. Elec nonamput | −10.6552 | −20.6829, −0.6274 | 0.0304 |
| Month 6 | | | |
| Group comparison | Difference between means | Simultaneous 95% confidence limits | *p*-value |
| Non-electrical amput vs. Non-electrical nonamput | −6.5665 | −12.4164, −0.7167 | 0.0185 |
| Elec amput vs. Elec nonamput | −7.6410 | −18.8590, 3.5769 | 0.4309 |
| Month 12 | | | |
| Group comparison | Difference between means | Simultaneous 95% confidence limits | *p*-value |
| Non-electrical amput vs. Non-electrical nonamput | −7.1189 | −12.6199, −1.6189 | 0.0040 |
| Elec amput vs. Elec nonamput | −7.5845 | −18.6486, 3.4795 | 0.4197 |

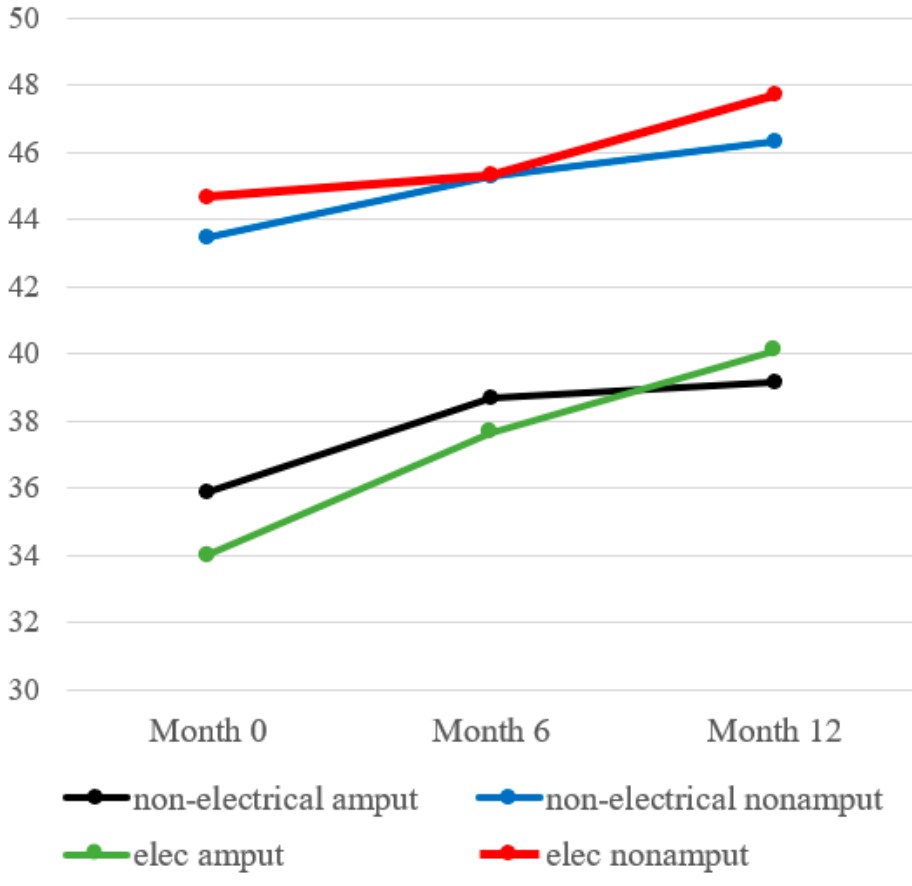

**Figure 2.** Trend of physical component score.

*3.4. Mental Component Score (MCS)*

The results are summarized in Table 4 and Figure 3. The MCSs were not significantly different across groups at discharge, 6 months, or 12 months post-burn.

**Table 4.** Mental component score (MCS) with Kruskal–Wallis test.

| | | Month 0 | | | | |
| --- | --- | --- | --- | --- | --- | --- |
| **Variable** | **Statistics** | **Non-electrical amput** | **Non-electrical nonamput** | **Elec amput** | **Elec nonamput** | *p*-value |
| MCS | Mean $\pm$ SD | 45.27 $\pm$ 13.80 | 49.52 $\pm$ 12.07 | 46.00 $\pm$ 14.27 | 51.21 $\pm$ 10.71 | 0.2511 |
| | Median | 45 | 51 | 46 | 51 | |
| | *n* | 30 | 516 | 11 | 29 | |
| | | **Month 6** | | | | |
| **Variable** | **Statistics** | **Non-electrical amput** | **Non-electrical nonamput** | **Elec amput** | **Elec nonamput** | *p*-value |
| MCS | Mean $\pm$ SD | 46.85 $\pm$ 11.15 | 50.83 $\pm$ 11.67 | 51.22 $\pm$ 10.00 | 52.46 $\pm$ 12.40 | 0.1426 |
| | Median | 48 | 53 | 50 | 56 | |
| | *n* | 26 | 452 | 9 | 26 | |
| | | **Month 12** | | | | |
| **Variable** | **Statistics** | **Non-electrical amput** | **Non-electrical nonamput** | **Elec amput** | **Elec nonamput** | *p*-value |
| MCS | Mean $\pm$ SD | 48.11 $\pm$ 13.34 | 50.96 $\pm$ 12.25 | 55.67 $\pm$ 10.40 | 50.26 $\pm$ 12.82 | 0.3622 |
| | Median | 48 | 55 | 60 | 54 | |
| | *n* | 28 | 400 | 9 | 23 | |

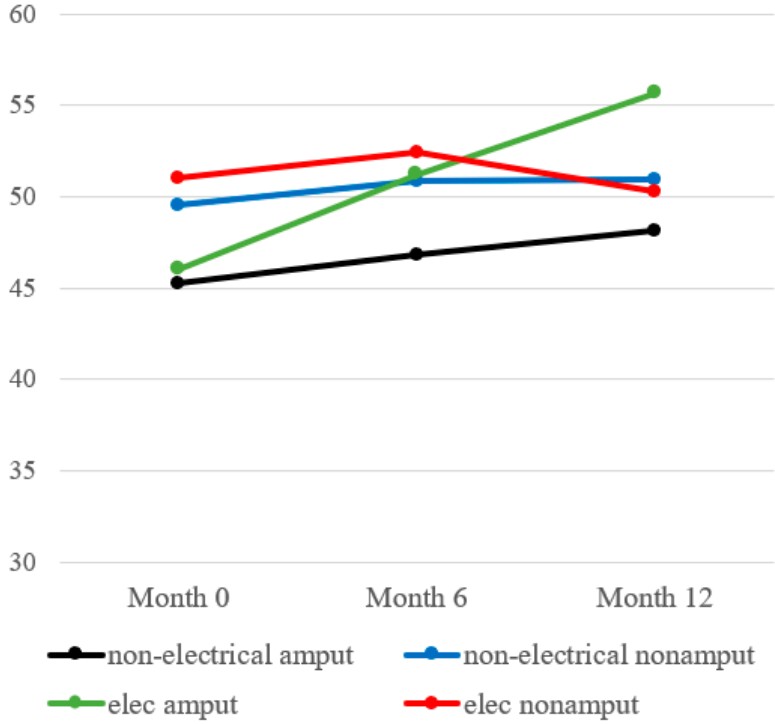

**Figure 3.** Trend of mental component score.

*3.5. Social Participation*

The social role results are summarized in Tables 5 and 6, and Figure 4. The Kruskal–Wallis test was significant ($p = 0.01$) at the time of discharge between the four study groups. Although the subsequent Dunn–Bonferroni post hoc test failed to demonstrate a significant difference between any study group, electrical burn patients who underwent amputation had a lower social role score than electrical burn survivors without amputation, which may be clinically significant (44.00 vs. 55.61; $p = 0.07$) at discharge. At one year post-injury, non-electrical burn survivors with amputation reported significantly lower social role score than non-electrical burn survivors without amputation (47.53 vs. 54.72; $p = 0.03$).

**Table 5.** Social role score with Kruskal–Wallis test.

| | | Month 0 | | | | |
|---|---|---|---|---|---|---|
| **Variable** | **Statistics** | **Non-electrical amput** | **Non-electrical nonamput** | **Elec amput** | **Elec nonamput** | ***p*-value** |
| Social Role Score | Mean ± SD | 48.26 ± 12.30 | 52.99 ± 10.74 | 44.00 ± 10.90 | 55.61 ± 11.29 | 0.0144 |
| | Median | 43 | 53 | 42 | 62 | |
| | *n* | 23 | 369 | 8 | 18 | |
| | | **Month 6** | | | | |
| **Variable** | **Statistics** | **Non-electrical amput** | **Non-electrical nonamput** | **Elec amput** | **Elec nonamput** | ***p*-value** |
| Social Role Score | Mean ± SD | 50.47 ± 9.78 | 54.52 ± 10.88 | 48.78 ± 10.05 | 54.67 ± 11.31 | 0.0772 |
| | Median | 51 | 59 | 49 | 59 | |
| | *n* | 19 | 355 | 9 | 21 | |
| | | **Month 12** | | | | |
| **Variable** | **Statistics** | **Non-electrical amput** | **Non-electrical nonamput** | **Elec amput** | **Elec nonamput** | ***p*-value** |
| Social Role Score | Mean ± SD | 47.53 ± 11.33 | 54.72 ± 10.16 | 55.71 ± 9.12 | 59.69 ± 9.76 | 0.0061 |
| | Median | 47 | 56 | 59 | 64 | |
| | *n* | 17 | 296 | 7 | 16 | |

**Table 6.** Social role score post hoc analysis with Dunn–Bonferroni test.

| | Month 0 | | |
|---|---|---|---|
| **Group comparison** | **Difference between means** | **Simultaneous 95% confidence limits** | ***p*-value** |
| Non-electrical amput vs. Non-electrical nonamput | −4.7337 | −10.9161, 1.4487 | 0.2581 |
| Elec amput vs. Elec nonamput | −11.6111 | −23.8346, 0.6124 | 0.0730 |
| | **Month 12** | | |
| **Group comparison** | **Difference between means** | **Simultaneous 95% confidence limits** | ***p*-value** |
| Non-electrical amput vs. Non-electrical nonamput | −7.1902 | −13.9307, −0.4496 | 0.0295 |
| Elec amput vs. Elec nonamput | −3.9732 | −16.2208, 8.2743 | 1.000 |

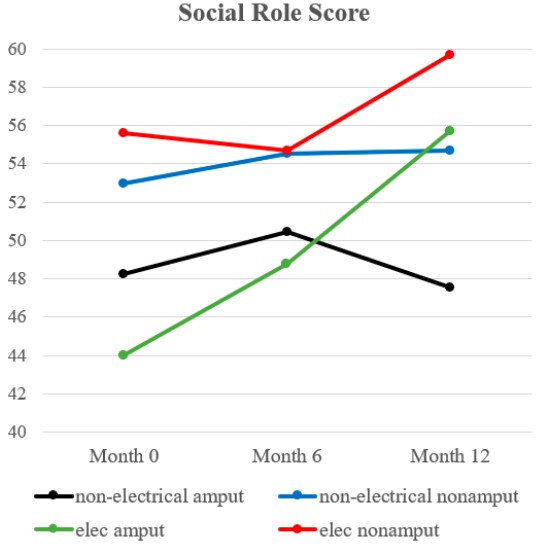

**Figure 4.** Trend of social role score.

## 4. Discussion

The first aim of the current study was to compare the incidence of amputation in patients with electrical burn injuries to individuals with different mechanisms of injury. The results of our study revealed that electrical burn patients had a significantly higher likelihood of undergoing amputation compared to non-electrical burn patients. Specifically, our study found an amputation rate of 30.3% among electrical burn patients, whereas the amputation rate among non-electrical burn patients was only 6.56%.

These findings are consistent with results from a 10-year retrospective analysis of a burn center in South China, which reported similar amputation rates of 37.33% in high-voltage injuries [25]. However, studies conducted in other countries have yielded different results. For example, a retrospective study in a specialized provincial burn center in Ontario, Canada, showed that only 13% of patients with electrical burns required amputation [26]. Another study conducted in the Zurich Burn Center over a 15-year period also reported an amputation rate of 13.5%, which aligns with the findings from Canada [27].

It is challenging to determine the exact reasons for the variation in amputation rates across different regions in electrical burn injuries. Several factors could contribute to this variation, including the prevalence of electrical burns in different countries. China, with its large population in urban areas and rapid industrialization with an extensive use of electrical infrastructure, has been reported to have a high rate of electrical burn accidents [28]. It is also worth noting that studies conducted in Zurich and Ontario focused on a single center, whereas our research utilizes a national database, which may provide more comprehensive and generalizable results [26,27]. This difference in study design, particularly national-scale studies, could shed light on the variations in the rates of electrical burn injuries by geographic region, such as urban or rural settings. Furthermore, the study in the Zurich Burn Center predicted that the development of compartment syndrome, rhabdomyolysis, high myoglobin and CK blood levels, kidney failure, sepsis, and respiratory complications during the hospital course were predictive of a higher amputation rate [27]. As such, factors that contribute to these conditions, such as access to quality healthcare, workplace safety policies, and the proportion of jobs involving high-voltage electrical hazards, in a particular country may also influence the incidence of amputation in electrical burn injuries.

The current study revealed higher amputation rates in electrical burn injuries in the United States compared to other Western countries. The high amputation rate is concerning and warrants further investigation. Some contributing factors may include differences in the types of employment or workplace safety practices between different countries. This discrepancy suggests the need for further investigation into the contributing factors in order to develop more effective prevention and treatment approaches.

The second aim of the current study was to examine physical and mental health outcomes following electrical injuries with and without amputation. The negative impact of amputation on physical functioning among patients with electrical burns was clearly observed in our study. We found that participants with electrical burns who underwent amputation reported significantly lower PCSs at discharge compared to those without amputation. Previous studies have shown that while patients with amputation gain some functionality back with prosthetic devices, a third to over half may experience dissatisfaction with aspects such as prosthetic limb comfort, residual limb skin health, or amputation related pain [29–31]. Other studies have identified various factors that negatively affect quality of life after amputation, including older age, cognitive impairment, presence of phantom pain and stump pain, the way patients move, decreased independence, diminished occupational activity, and limited access to rehabilitation [32,33].

Previous studies have indicated that individuals with bilateral or more proximal amputations tend to experience more severe impairments in physical capacity and walking ability [34,35]. One limitation of the current study was the lack of data regarding the specific location, extent, and details of the amputation in the database, which could have provided important additional information about outcomes. Taken together, the current

study and previous research highlight the ongoing challenges faced by individuals living with amputation and the need to understand multifaceted factors that can help guide healthcare providers in developing personalized approaches to patients with amputation. The study results also reinforce the importance of understanding the factors that can significantly impact the quality of life for patients with amputations and underscore the importance of initiating comprehensive physical rehabilitation early in recovery to assist with regaining physical function.

Interestingly, our study also revealed that the adverse effects of amputation were more prolonged in patients with non-electrical burns than in patients with electrical burns. Specifically, the electrical burn group with and without amputation did not experience significant differences in their PCS or social role score at six months post-injury. In contrast, non-electrical burn survivors with amputation faced more persistent challenges in physical function and produced lower social role scores up to one year post-injury. These results align with a study focusing on pediatric patients with electrical burn injuries, which demonstrated that although electrical burn patients had more extensive limb loss and major amputations, their hospital stay was approximately two weeks shorter than that of non-electrical burn patients [36].

One possible explanation for the shorter-term impact of amputation on electrical bur n patients is their lower TBSA (%) burnt compared to non-electrical burn patients. A greater TBSA (%) burnt is known to contribute to higher mortality, comorbidities, and longer lengths of stay in the hospital [37–39]. Amputation may have more immediate impact on physical and social functioning, while patients with large TBSA (%) burnt injuries often require frequent wound care, multiple surgeries, scar therapies, and rehabilitation, in addition to dealing with the acute amputation. These ongoing stressors may contribute to the longer-lasting and more severe physical and social difficulties experienced by non-electrical burn patients with amputation even up to one year post-injury.

The results of this study did not reveal significant differences in the mental component scores between electrical burn patients with and without amputation, suggesting that limb differences may not be associated with additional psychological distress. Previous studies have found profound long-term psychological stress and psychiatric disorders in electrical burn patients, including body dysmorphic disorder, depression, anxiety, and PTSD [40,41]. There is also evidence of significantly higher rates of insomnia in trauma survivors, including those with burn injuries [42]. Our study is consistent with previous research demonstrating that burn survivors should be closely monitored for the development of psychological disorders. These results also highlight the importance of providing robust psychological and social support during all phases of recovery from burn injury, regardless of mechanism or amputation status.

Based on our findings, it is evident that there is a need for in-depth research on effective rehabilitation plans for patients with electrical burns and amputation, beginning in the acute phase and continuing through discharge and recovery. Implementing more intensive and comprehensive interventions during the early stages of rehabilitation can lead to better long-term physical, psychological, and social outcomes for electrical burn survivors with amputation. Conversely, rehabilitation plans for non-electrical burn patients with amputation should focus more on long-term management for at least one year post-injury due to the significantly longer and more severe difficulties they face, particularly considering the larger TBSA (%) burnt.

Moreover, this study's findings reinforce the importance of developing better work-place electrical injury prevention and rehabilitation programs, particularly for Hispanic individuals. Although the study was conducted in the context of the US healthcare system, which may limit generalizability, the differences in the ethnicity characteristics in this study sample are striking and potentially indicative of underlying disparities with broad applicability. Past studies have confirmed the presence of racial disparities in intensive rehabilitation for the Hispanic population [43,44]. Researchers found that Hispanic patients were less likely to receive a higher level of rehabilitation, regardless of insurance

coverage or prehospital characteristics. Further education on health care disparities in the Hispanic population is crucial, as this group has the greatest risk factor for electrical burn injuries while having a lower chance of receiving a higher level of rehabilitation services among burn-injured patients [43,44]. Addressing these disparities requires a multi-faceted approach to enhance workplace safety measures and culturally sensitive rehabilitation programs. Efforts should be made to raise awareness among healthcare providers about the specific risks of injury and needs of Hispanic patients and to promote equitable access to rehabilitation services. The American healthcare system should further effectively address the unique challenges faced by Hispanic patients in the realm of electrical injury prevention and rehabilitation.

Although the current study includes a large amount of data from a longitudinal multi-site project, it does have limitations. The study data was retrieved from the Burn Model System National Database, which is limited to the variables selected by the participating centers since the project began. As such, the data did not include several pertinent variables for participants after 2005, such as the location and extent of amputation, which may have allowed for a more nuanced understanding of post-amputation outcomes. Future studies may include additional data regarding medical complications, such as length of stay, ICU admission, intubation, and the exact role of TBSA (%) burnt in long-term outcomes for electrical burn patients. It will also be important to study social and cultural factors that can influence treatment and outcomes, such as income, language, and family support.

## 5. Conclusions

In conclusion, patients with electrical burns in this large multi-site study had a higher amputation rate (30.3%) than patients with non-electrical burns (6.6%). Patients with electrical burn and amputation experience significant physical and social challenges at discharge and may benefit from early rehabilitation and support interventions. The higher amputation rate among patients with electrical burns highlights the importance of prioritizing preventive measures and safety protocols in workplaces to reduce the occurrence of these devastating injuries. Additionally, a significant proportion of individuals with electrical injuries were Hispanic, suggesting the need for both tailored injury prevention efforts and post injury care to specifically meet the needs of this population. Patients with non-electrical burns with amputation require greater long-term monitoring of physical and mental changes, as this population seems to experience more prolonged challenges than those with electrical injuries regardless of amputation due to a larger TBSA (%) burnt. Overall, comprehensive mental support is essential for all burn patients, irrespective of burn etiology or amputation status, to facilitate their successful reintegration into daily life.

**Author Contributions:** Conceptualization, K.K.; Methodology, B.W.; Formal analysis, B.W. and K.K.; Investigation, E.K., B.W. and K.K.; Data curation, K.J.S.-B.; Writing—original draft E.K.; Writing—review & editing, E.K. and K.K.; Supervision, K.J.S.-B. and K.K.; Project administration, K.J.S.-B. and K.K. All authors have read and agreed to the published version of the manuscript.

**Funding:** The contents of this manuscript were developed under a grant from the National Institute on Disability, Independent Living, and Rehabilitation Research (NIDILRR grant number 90DPBU0006). NIDILRR is a Center within the Administration for Community Living (ACL), Department of Health and Human Services (HHS). The contents of this manuscript do not necessarily represent the policy of NIDILRR, ACL, or HHS, and you should not assume endorsement by the Federal Government.

**Institutional Review Board Statement:** This study was exempt from the institutional review board approval because the information obtained from the database was a nationally available de-identified dataset.

**Informed Consent Statement:** Not applicable.

**Data Availability Statement:** The data are available in Burn Model System National Database.

**Conflicts of Interest:** The authors declare no conflict of interest.

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
