# Peer review of "Outcomes of Patients with Amputation following Electrical Burn Injuries"

_2673-1991, doi:10.3390/ebj4030029_

Round 1
Reviewer 1 Report
The authors report results from a national database about the physical, mental and social impact of amputation on electrical and non-electrical burn injury. I find the issue important and I believe it will be interesting for readers.
I have a few comments that I believe will help improve the manuscript.
1. Methods - it is disappointing that the level or anatomic region of amputation does not exist in the database. I believe comparison of this variable between the groups could have shed additional light on the results. Are the authors positive there is no way of adding this data?
2. Do the authors have a letter of exemption from an IRB?
3. Results – suggest to write mean±SD instead of the long sentence in lines 100-101.
4. Please add a patient baseline characteristics table according to the different study groups with relevant p values, include age, gender, ethnicity, TBSA, and additional baseline variables if you have them.
5. Please add n (%) to table 1.
6. Were both electrical groups PCS results significantly lower than both non-electrical groups results?
7. Control amput, control nonamput etc. – these groups are discussed in the text as “non-electrical” groups, not control groups. Please choose at your preference and use the same definitions in the main text, tables and figures. Personally, I think than non-electrical is clearer.
8. Line 143 p-value of 0.0144 – I think this can be rounded to 0.01.
9. Discussion - The authors discuss the fact that there is need for better workplace electrical injury prevention and rehabilitation program for those who speak Spanish, as the vast majority of these patients were Hispanic. The authors then compare their results to other studies. This is excellent, but this is the only result that is compared to other studies in the Discussion section. The same should be done for the other (main) results of this study, for instance comparison to the “Previous research (references 5,15-17)” mentioned in the Introduction section.
10. Additionally consider mentioning/discussing these additional studies as well:
11. Pedrazzi N, Klein H, Gentzsch T, Kim BS, Waldner M, Giovanoli P, Plock J, Schweizer R. Predictors for limb amputation and reconstructive management in electrical injuries. Burns. 2023 Aug;49(5):1103-1112.
12. Kennedy PJ, Young WM, Deva AK, Haertsch PA. Burns and amputations: a 24-year experience. J Burn Care Res. 2006 Mar-Apr;27(2):183-8.
13. Tapking C, Hundeshagen G, Popp D, Lee JO, Herndon DN, Zapata-Sirvent R, Branski LK. The Frequency and Reason For Amputations in Electrically Burned Pediatric Patients. J Burn Care Res. 2019 Jan 1;40(1):107-111.
Minor mistakes, please correct.
Reviewer 2 Report
Title
‘ Outcomes of amputation following electrical burn injuries’
- I believe that the title should be amended to include a phrase about the outcome of patients with amputation, rather than the outcome of the amputation itself.
Abstract
‘Electrical burn patients with amputation showed significantly lower physical component scores (PCS=34.00) at discharge than electrical burn patients without amputation (p < 0.05).’
- Adding PCS for electrical burn patients without amputations would be helpful.
Introduction
‘Mortality rates from electrical 28 injuries at high voltages can be as high as 5.2%.
Moreover, electrical injuries pose unique challenges compared with burns from other 32 etiologies, as the extent of injury can be difficult to assess at initial presentation’
‘Burn survivors with amputations tend to experience a severe social stigma and require intense targeted counseling and social support.’
- References should be added
Material and Method section
The overall mean difference in the PROMIS29 and VR12 scores between the patient groups were analyzed using Kruskal-Walli’s test and post-hoc analysis using Dunn-Bonferroni test.
This test should be referred to as Kruskal–Wallis test. Correct it in various points in the text, legends and tables.
Results
The median age of patients admitted for burns was 30.18 y
- I would prefer that you refer to age without abbreviations.
3.1. Demographics
‘The Hispanic 104 population comprised the largest percentage of the electrical burn group (80%). The 105 mean age of Hispanic population with electrical burn injury was 37.15 y, median age of 106 36.85 y, with a standard deviation of 14.11 y.’
- It would be helpful if the authors could provide corresponding information for non-Hispanics.
- It would be interesting if the authors could provide similar information for non-electrical burns.
-I would recommend creating a new table with 3 columns including separately depicted data of all patients, data of patients with electrical injury and data of patients without electrical injury with corresponding p values.
Table 1.
- I suggest omitting the Table 1 and presenting the data only in the text form.
Table 2 , pcs column 1
- pcs should be written in capital letters- as PSC . Explanation of abbreviations should be provided in the table’s legend/sub notes.
Table 4, first column, mcs- the same suggestion – use the capital letters - MCS . Explanation of abbreviations should be provided in the table’s legend/sub notes.
Discussion section.
In my opinion, the discussion is the weakest point of the article. It should be rewritten, references to the existing literature should be added and their results should be compared with the results of other studies.
Discussion section, first paragraph.
- It is not clear whether the work focuses on patients with burns or patients with any other trauma. I would suggest that the discussion begins with clear reference to burn patients.
Sixth paragraph, last sentence …Further education on health care disparities in the Hispanic population is crucial, as this group has the greatest risk factor for electrical burn injuries while having a 200 lower chance of receiving higher level of rehabilitation services among burn-injured patients.
- References should be provided.
no comments
Reviewer 3 Report
1. Pg 2, line 48: I suggest changing "inevitable" to "mandatory"
2. As referred, this study, in spite its great scientific quality, is reletively biased paying attention to the population sample and to the specific conditions of the North-American Health System. I my opinion, this fact sgould be more stressed and it would be interesting to compare the results with similar data from other countries, namely in Europe, Canada or China, for instance.
Round 2
Reviewer 2 Report
Several minor flaws, listed below, could possibly be improved on by minor revision.
Tab 1. 1st column TBSA burn (%), text and legends.
- I would suggest abbreviating TBSA burn assessment as TBSA (%) burnt in the text, legends and tables.
Fig. 2 and Fig. 4
non-electrical nonamput, elec non amput
- Use the same formatting for this abbreviation: ‘non amput’or ‘nonamput’
Appropriate quality of language, minor remarks
